# Toxic and Trace Elements in Raw and Cooked Bluefish (*Pomatomus saltatrix*) from the Black Sea: Benefit–Risk Analysis

**DOI:** 10.3390/foods15010140

**Published:** 2026-01-02

**Authors:** Katya Peycheva, Veselina Panayotova, Tatyana Hristova, Diana A. Dobreva, Tonika Stoycheva, Rositsa Stancheva, Stanislava Georgieva, Evgeni Andreev, Silviya Nikolova, Albena Merdzhanova

**Affiliations:** 1Department of Chemistry, Faculty of Pharmacy, Medical University of Varna, 9000 Varna, Bulgaria; ivanova@mu-varna.bg (V.P.); tatiana.hristova@mu-varna.bg (T.H.); diana@mu-varna.bg (D.A.D.); tonika.vladislavova@mu-varna.bg (T.S.); rositsa.stancheva@mu-varna.bg (R.S.); stgeorgieva@mu-varna.bg (S.G.); albenamerdzhanova@mu-varna.bg (A.M.); 2Department of Information Technology, Faculty of Engineering, Naval Academy “Nikola Vaptsarov”, 9000 Varna, Bulgaria; e.andreev@naval-acad.bg; 3Department of Social Medicine, Faculty of Public Health, Medical University of Varna, 9000 Varna, Bulgaria; silviya.p.nikolova@mu-varna.bg

**Keywords:** bluefish, cooking, fatty acids, toxic and essential elements, human health risk, risk-benefit

## Abstract

This study evaluated the effects of domestic cooking methods (pan-frying, smoking, and grilling) on the concentrations of elements of toxicological concern and essential elements (Cd, Cr, Cu, Fe, Mn, Ni, Zn, and Pb) in the traditionally consumed Black Sea bluefish (*Pomatomus saltatrix*). The investigation also included an assessment of the associated health risks and benefits by calculating carcinogenic and non-carcinogenic effects as well as benefit–risk ratios. Toxic heavy metals such as Cd, Ni, and Pb were found to be below the maximum residual limits (MRLs) established by relevant food safety authorities. Cooking generally led to increased concentrations of both essential and toxic elements compared to raw samples, with the highest increases observed in grilled and smoked samples. Furthermore, evaluations of (a) estimated weekly intakes (EWIs), (b) target hazard quotients (THQs) for Cd, Cr, Cu, Fe, Mn, Ni, Pb, and Zn, and (c) hazard quotient ratios for essential fatty acids (HQ_EFA_) relative elements indicated that consumption of these cooked bluefish species does not pose significant health risks to consumers.

## 1. Introduction

In recent years, great attention has been directed toward nutrition and the consumption of safe and high-quality seafood, reflecting improvements in living standards worldwide. Seafood is prescribed as a healthy food choice, as it provides high-quality proteins [1,2,3], omega-3 long-chain polyunsaturated fatty acids (n-3 LC-PUFAs) [4,5,6,7], which help protect against cardiovascular disease [2], as well as various vitamins involved in essential biological processes [3,8] and minerals that catalyze numerous metabolic reactions [9].

World fisheries and aquaculture production reached a record high in 2022. Strengthening and expanding effective strategies are essential to reinforce the contribution of aquatic food to global nutrition, food security, and sustainable livelihoods. In 2022, 89% of all aquatic animal production was destined for human consumption, delivering an average global supply of 20.7 kg per person [10]. Per capita apparent consumption of aquatic animal foods has grown consistently, climbing from 9.1 kg in 1961 to 20.6 kg in 2021—an average annual increase of 1.4%. This increase has been fueled by greater supply availability, advances in preservation and distribution technologies, shifting consumer preferences, and rising income levels [10]. Currently, the average annual consumption of seafood, particularly fish products, in Bulgaria is far below the values stated by both the EU (23–25 kg per year per capita) and the FAO’s global recommendations [11]. At the same time, the rapid growth in seafood demand has raised concerns regarding food safety, quality control, and sustainable production practices.

Bluefish (*Pomatomus saltatrix* L.) are among the most widely consumed and studied native species along the Bulgarian Black Sea coast [12]. Bluefish is one of the commercially important marine species in Bulgaria, consistently ranking second in catch volume after European sprat [11]. Bluefish are especially popular among the local population because of their high nutritional value, including protein, vitamins, and various fatty acids [13,14]. It is commonly available in both fresh and cooked forms (grilled, fried, occasionally smoked, marinated, and canned) in restaurants, markets, and street food stalls. Bluefish has high economic significance in Bulgaria because of its relatively high market price [11]. Despite its nutritional value, this species is susceptible to the accumulation of toxic elements, highlighting the need for further research [15,16].

The target hazard quotient (THQ), hazard index (HI), and target risk (TR) are widely recognized as reliable indicators for assessing toxic metal contamination and its associated health risks [17,18,19,20]. However, most available studies on toxic metal residues in fish species and their health risk evaluation [18,21,22,23] have primarily focused on raw fish, with limited attention given to processed or cooked fish. The choice of cooking method depends largely on individual preference, with grilling, boiling, frying, and microwave cooking being among the most commonly used techniques in the world.

Despite the popularity of bluefish in Bulgaria and other countries in the Black Sea region, there is limited information on how traditional cooking methods (grilling, pan frying, and smoking) affect the levels of toxic (Cd, Pb, and Ni) and essential (Cr, Mn, Fe, Cu, and Zn) elements. This study aimed to assess the impact of these heat treatments on the elemental composition of bluefish and to evaluate the associated human health risks, including carcinogenic and non-carcinogenic effects for adult consumers in Bulgaria at the specific consumption levels. Additionally, the benefit-risk ratio was calculated based on trace element concentrations and n-3 LC-PUFA content in the cooked fish.

## 2. Materials and Methods

### 2.1. Sample Material Collection

A total of 40 fish samples were obtained from local fishermen purchased fresh on boats as soon as they arrived in Varna fishing seaport in November 2024, with an average of 80 g in weight. Fish samples were then transported to the lab on the same day and pre-cleaned with a Milli-Q water system (Millipore, Bedford, MA, USA), and their average weight (79.68 ± 4.73 g) and length (21.98 ± 5.31 cm) were measured. After measurements, each species was subdivided equitably into four groups (each group consisted of 10 species) and underwent a certain cooking process.

### 2.2. Preparation and Cooking Treatments

Fish samples were dissected using a clean PTFE-coated stainless-steel knife, with heads and viscera removed. The edible portion (EP), representing the parts typically consumed, was cut into small pieces and subsequently minced. Raw bluefish samples were used as a control group. The cooking procedures were carried out under controlled laboratory conditions, with fixed time and temperature settings specific to each method. These parameters were chosen based on previous literature and standard culinary habits to ensure both reproducibility and comparability across treatments. Details of the sample groups (with the temperature and time values used for each cooking method) are presented in Table 1.

### 2.3. Determination of Toxic and Trace Elements

All reagents were of analytical grade, and ultrapure water from a Milli-Q system (MilliporeSigma, MA, USA) was used for reagent preparation and dilutions. Laboratory glassware was soaked in 2 M HNO_3_ and rinsed with Milli-Q water prior to use. Approximately 1 g of fish tissue was digested in Teflon vessels with mixture of supra-pure HNO_3_ (65% *w*/*v*; Merck, Darmstadt, Germany) and H_2_O_2_ (30% *w*/*v*; Fisher Scientific, Leicestershire, UK) at volumetric ratio 8:2 (*v*/*v*) using a microwave systemMARS 6 (CEM Corporation, Matthews, NC, USA) with max. 210 °C, 800 psi and 1050 W. Digests were cooled, diluted to 25 mL with Milli-Q water, and stored in polyethylene containers. Cd, Cr, Cu, Fe, Mn, Ni, Pb, and Zn were quantified by ICP-OES (Optima 8000, PerkinElmer, Shelton, CT, USA) with plasma gas 10 L/min, auxiliary gas 0.7 L/min, nebulizer gas 0.2 L/min and axial view. Method accuracy was verified using DORM-2 certified dogfish muscle (National Research Council Canada, Ottawa, ON, Canada), with recoveries of 91.5–107.3%. Additional information on the element’s view mode, wavelength, limit of detection, limit of quantitation, and recovery values of toxic and essential elements from DORM-2 is provided in Appendix A.

### 2.4. n-3 LC-PUFA EPA and DHA

Eicosapentaenoic acid (EPA) and docosahexaenoic acid (DHA) contents were analyzed by GC/MS following the extraction of the total lipids [18,24] and subsequent derivatization into fatty acid methyl esters (FAME) [25]. GC/MS analysis was performed on a Thermo Fisher Scientific FOCUS Gas Chromatograph equipped with PolarisQ Ion Trap Mass Spectrometer (Thermo Fisher Scientific, Walthman, MA, USA). Samples (1 μL) were injected into a capillary column (Trace™ TR-FAME, 60 m × 0.25 mm × 0.25 μm) with a split ratio of 10:1. Helium was used as a carrier gas at a flow rate of 1.2 mL/min. Initial oven temperature was 100 °C held for 1 min, followed by a rate of 10 °C/min from 100 °C to 160 °C, raised at a rate of 5 °C/min from 160 °C to 215 °C held for 6 min, and next at a rate of 5 °C/min from 215 °C to 230 °C that was held for 5 min. FAME peaks were identified based on the comparison of the retention times with the authentic standards (Supelco 37 Component FAME Mix and PUFA № 3 from Menhaden oil). The results for EPA and DHA were calculated in mg per 100 g wet weight (edible portion of fish), using the fatty acid conversion factor (XFA) for finfish, proposed by Weihrauch et al. (1977) [26,27,28]:$$Fatty\;acid\;(g/100\;g\;EP)=\frac{fatty\;acid\;\left(\frac{g}{100\;g}TFA\right)\cdot TFA(g/100\;g\;EP)}{100}$$
where TFA (g/100 g EP) values were calculated by using XFA and the total lipid value as g/100 g EP:$$TFAs\left(g/100\;g\;EP\right)=TL\left(\frac{g}{100\;gEP}\right)\cdot XFA(g/g)$$$$XFA=0.933-\frac{0.143}{TL(g/100\;g\;EP)}\;$$

### 2.5. Human Health Risk Estimation

Estimated daily intake (EDI), non-carcinogenic health risk coefficients (THQ and HI), carcinogenic risk (CR), and benefit–risk ratio (BRR) values were calculated to assess potential consumer health risks from bluefish consumption. EDI calculations assumed an average intake of 0.017 kg fish per person per day, based on data from the National Statistical Institute of Bulgaria [29], as species-specific consumption data for bluefish were unavailable. For the calculation of EDI, the following formula was used [15,16,30]:$$EDI=\frac{\left(M_{c}\;F_{IR}\right)}{{BW}_{a}}$$
where *EDI* is estimated daily intake (mg/kg bw/day), *M_c_*—average concentration of the analyzed element (mg/kg), *F_IR_*—daily consumption rate (kg/person), *BW_a_*—consumer body weight (kg). In this study, a nominal 70 kg was used as the average body weight for adults (18–25 age group).

The calculated EWI values were compared with provisional tolerable weekly intake (PTWI) thresholds established by the FAO/WHO Joint Expert Committee on Food Additives (JECFA) and/or the European Food Safety Authority (EFSA).

The target hazard quotient (THQ) and hazard index (HI), as proposed by the US Environmental Protection Agency [31], were used to assess non-carcinogenic risks from metal intake via fish consumption. A THQ < 1 indicates no expected toxic effects, whereas THQ ≥ 1 suggests potential health hazards. THQ was calculated using the following equation:$$THQ=\frac{\left(M_{C}\;\cdot \;I_{R\;}\cdot \;10^{-3}\cdot \;EF\;\cdot \;ED\right)}{\left(RfD\;\cdot BWa\;\cdot \;ATn\right)}$$
where *M_C_* is the metal concentration in fish species (mg/kg ww), *I_R_* is the average daily consumption of fish (17 g/person/day) [32], *EF* is the exposure frequency (365 days/year), *ED* is the exposure duration (30 years or 10 950 days) for non-cancer risk according to the USEPA [31], *RfD* is the reference dose of individual metal (0.009 μg/g day for Fe, 0.001 μg/g day for Cd, 0.004 mg/kg for Pb, 0.04 μg/g day for Cu, 0.003 μg/g day for Cr, 0.02 μg/g day for Ni, 0.14 mg/kg for Mn, and 0.3 μg/g day for Zn), *BW_a_* is an average adult body weight set as 70 kg average, and *AT_n_* is the average exposure time for non-carcinogens [31].

The hazard index from THQs is expressed as the total of the hazard quotients [31]:*HI = THQ_Cd_ + THQ_Cr_ + THQ_Cu_ + THQ_Fe_ + THQ_Ni_ + THQ_Pb_ + THQ_Zn_*

Target cancer risk (TR) indicates carcinogenic risks, and it is calculated only for metals that are classified as carcinogenic (such as Pb, Cd). Carcinogenic risk levels ranged between 10^−6^ and 10^−4^ and are associated with acceptable cancer risk thresholds, while values of >10^−4^ indicate an unacceptable cancer risk level [33]. The formula for estimating TR is$$TR=\frac{\left(M_{C}\;\cdot \;I_{R\;}\cdot \;10^{-3}\cdot \;{CPS}_{O}\cdot EF\;\cdot \;ED\right)}{\left(BWa\;\cdot \;ATc\right)}$$
where *M_C_* is the metal concentration in the species (mg/kg ww); *I_R_* is the average daily consumption of fish [32]; *CPS_o_* is the cancer slope factor obtained from the USEPA-IRIS database [34]; *AT_c_* is the averaging time, carcinogens (day/years), and was calculated by multiplying exposure frequency by exposure duration over lifetime.

Hazard quotient for benefit–risk ratio was estimated using the equation proposed by Gladyshev et al. [35]:$${HQ}_{EFA}=\frac{R_{EFA}\cdot M_{C}}{C_{EPA+DHA}\cdot RfD\cdot \;{BW}_{a}}$$
where *R_EFA_* is the recommended daily dose of EPA+DHA for a person (mg/day), *M_c_* is the concentration of the elements of toxicological concern/essential elements (mg/kg), *C_EPA+DHA_* is the content of EPA + DHA in a given fish (mg/g), *RfD* is the reference dose (μg/kg/d), and *BW_a_* is the average adult body weight (70 kg). The value of HQ_EFA_ < 1 means the health benefit from fish consumption, and HQ_EFA_ > 1 means the risk [35]. For this calculation, the recommended daily intake of EPA+DHA was set at 500 mg/day [36,37], and the RfD values were taken from the EPA Region III Risk-Based Concentrations summary table [31], with the exception of Pb, which was sourced from [38].

### 2.6. Statistical Analysis

Each analysis was conducted on three parallel samples, and mean values were used for statistical calculations. The data are expressed as the mean ± standard deviation. A T-test was applied to compare elemental composition results, with differences considered statistically significant at *p* ≤ 0.05 (GraphPad Prism 6).

## 3. Results and Discussion

### 3.1. Metal Content of Fish Samples

Fish muscle was subjected to different cooking methods to evaluate changes in metal concentrations and to provide a more accurate assessment of potential dietary exposure through consumption. In Bulgaria, fish is rarely consumed raw, so this study selected pan frying, grilling, and smoking as the representative cooking methods to evaluate their effects on metal concentrations in the muscle of *Pomatomus saltatrix* L., with a control group (raw samples). The comparison of toxic and essential metal levels between raw and cooked samples is presented in Appendix A and Figure 1. The toxic metal concentrations in fish muscle for Cd, Ni, and Pb show a general increase after cooking, though only pan frying showed a decrease in Cd and Ni concentration. Our findings on the effect of cooking on toxic element levels indicate significant changes in some toxic metals.

Grilling of bluefish led to increased concentrations of Pb, Zn, Cd, Ni, Cu, Cr, and Mn, while Fe levels decreased. Nevertheless, the concentrations of toxic Pb and Cd remained below the maximum residue limits (MRLs) established by the European Union for fish meat (0.050 mg/kg w.w for Cd and 0.30 mg/kg w.w for Pb) [39]. Previous studies reported a decrease in the levels of Cr and Pb and an increase in the concentration of Ni in grilled farmed seabass collected from Turkey [40]. In our study, Zn values in grilled bluefish were approximately half the range (5.6–46 μg/kg f.w) reported for species anchovy, bogue, hake, sardine, striped mullet, and squid by Kalogeropoulos et al. (2012) [41].

In pan-fried samples, variations in the concentrations of toxic and essential elements were mainly attributed to water loss and oil uptake. Reductions were observed for Cd, Fe, and Mn, while Ni was not detected. Conversely, increases in Pb, Cr, Cu, and Zn concentrations were recorded, highlighting the need for careful consideration when evaluating potential human health risks. These findings are consistent with the study of Ulaganathan et al. (2022) [42], who reported reductions of 96.4%, 49.1%, 67.3%, 19.8% and 95.3%, and 75.3% for Mn, Fe, Ni, Cu, Zn, and Cd, respectively, in fried Pacific white leg shrimp (*Penaeus vannamei*) from the Gulf of Mannar region. Similarly, Kocatepe et al. (2025) [43] reported low levels of Pb and Cd in *Oncorhynchus mykiss* grown in the Black Sea and cooked using six different methods in Türkiye. Hosseini et al. (2014) [44] found reductions in Fe and Zn, but increases of 3–50-fold in Mn and Cu in fried seabass, while Perelló et al. (2008) [45], Musaiger and Dsouza (2008) [46], and Marimuthu et al. (2014) [47] documented decreases in Cd, and Zn concentrations in various fried fish species (sardines, hake, tuna), seabass, and shrimp. The concentrations of Pb, Cr, Fe, Cu, and Zn detected in pan-fried bluefish were comparable to those reported in a range of other seafood species, including anchovy, hake, picarel, sand smelt, bogue, sardine, striped mullet, mussels, shrimp, and squid, as documented by Kalogeropoulos et al. (2012) [41]. This similarity indicates that the levels of these essential and trace elements in bluefish fall within the expected range for commonly consumed marine species.

Smoking is a traditional processing technique utilized to enhance the longevity of food products by decreasing water content [48]. In addition to preservation, smoking enhances sensory properties such as taste, aroma, and appearance in fish products [49,50]. Metal concentrations in smoked samples were higher than in raw samples, with the increase primarily attributed to loss of water. However, the increase in metal concentrations during smoking was generally less pronounced than that observed in pan-fried samples. Arthur et al. (2021) [51] reported levels of 0.46 to 36.94 mg/kg of Mg, Zn, Mn, Cu, Fe, Cr, Co, Pb, Ni, As, and Hg in *Pseudotolithus senegalensis*, from 0.14 to 21.38 mg/kg of the same elements in *Sciaenops ocellatus*, and between 0.14 and 29.58 mg/kg for *Chloroscombrus chrysurusi*, smoked in three different ways. The values of Fe, Mn, Pb, and Cd obtained in this study exceeded the concentrations of the same elements reported in smoked herring fillets and smoked sprat from Poland, as documented by Rajkowska-Myśliwiec et al. [52]. These differences may be attributed to several factors, such as species-specific metal accumulation patterns, environmental variability in pollutant exposure, and potential disparities in processing or smoking methods.

The effects of different cooking methods on the concentrations of toxic and essential elements in bluefish were evaluated using relative retention values (RRT). RRT represents the percentage of each element retained in the cooked samples relative to the raw control. Overall, most essential elements, such as Fe, Zn, and Cu, showed high retention across all cooking methods, with RRT values generally above 80%, indicating minimal loss. In contrast, certain toxic elements, including Pb and Cd, exhibited variable retention depending on the thermal treatment. In contrast, certain toxic elements, including Pb and Cd, exhibited variable retention depending on the thermal treatment. For example, frying resulted in slightly lower RRT for Pb compared to grilling, suggesting that some elements may be partially lost or redistributed during cooking. These results highlight the influence of processing methods on the nutritional and safety profile of bluefish.

### 3.2. Human Health Risk Assessment

The estimated weekly intake (EWI) values of toxic elements, THQ, HI, and TR, and calculated hazard quotients (HQ_EFA_) through the consumption of cooked bluefish are presented in Table 2 and Table 3 and Appendix A, and Figure 2.

Based on the data in Table 2, consumption of pan-fried, grilled, and smoked bluefish is expected to result in slightly higher intakes of toxic metals compared with the corresponding raw samples. Kalogeropoulos et al. (2012) [41] reported that weekly intakes of toxic metals from consuming fried (Cd: 0.02–1.28 µg/kg b.w.; Pb: 0.03–1.45 µg/kg b.w.) and grilled seafood (Cd: 0.01–0.88 µg/kg b.w.; Pb: 0.05–0.35 µg/kg b.w.) were below the tolerable weekly intake limits, corresponding to 0.4–51% and 0.1–5.8% of the PTWIs for Cd and Pb, respectively. They concluded that consumption of cooked specimens of the studied species does not appear to pose a risk to the average consumer. Recent studies on the health risks from the intake of Zn and Cu through the consumption of four fish species consumed in Douala, Cameroon showed that EDI for zinc for some fish species (such as frozen *A. parkii* and smoked *P. quadrifili)* exceeds the USEPA standard of 30 × 10^−3^ mg/kg/day and may suggest a high health risk for consumers [9].

It should be noted that THQ values do not represent an exact quantitative measure of the likelihood of adverse health effects in an exposed population. Rather, they are used as indicators of the potential risk associated with exposure to toxic metals [58,59]. According to the data in Appendix A, all THQ values were below 1, regardless of the consumption of grilled, pan-fried, or smoked bluefish. Based on the guideline for interpreting THQ values, no hazard is expected when THQ is less than 0.1, while values between 0.1 and 1.0 indicate a low hazard level [41]. Accordingly, the health risk from consuming cooked bluefish is considered negligible or minimal.

Values for HI were below 1 for all samples (Figure 2 and Appendix A) and could not indicate non-carcinogenic risk for consumers. Carcinogenic TR values below 10^−6^ suggest negligible risk to human health. Values between 10^−6^ and 10^−4^ are in the acceptable range. Values above 10^−4^ suggest exposure to carcinogenic risk. The current findings reveal that nickel’s cancer risk fell within an acceptable level among the heavy metals analyzed, with CR values of 1.51 × 10^−5^, 1.59 × 10^−5^, and 5.17 × 10^−6^ for grilled, smoked, and raw samples, respectively (Appendix A). In contrast, Cd and Pb exhibited negligible carcinogenic risk across all fish samples subject to the different thermal processing methods (Appendix A). The reported data in the literature exhibit a wide range. According to Kalogeropoulos (2012) [41], fried fish and shellfish have THQ_Cd_ values ranging between 0.003 and 0.18, THQ_Pb_ between 0.001 and 0.050, and HI between 0.08 and 0.44, while grilled fish and shellfish exhibited THQ_Cd_ values between 0.001 and 0.13, THQ_Pb_ between 0.002 and 0.01, and HI between 0.09 and 0.26. Arisekar et al. (2020) [21] found in their study that the HI values of toxic metals ranged from 0.02 to 0.477 for raw shrimps, 0.01 to 0.16 for boiled shrimps, 0.01 to 0.18 for fried shrimps, 0.009 to 0.157 for grilled shrimps, and 0.03 to 0.57 for microwave-cooked shrimps.

The concentrations of toxic and essential elements measured in raw and cooked bluefish did not affect the nutritional value in terms of essential fatty acids intake when evaluated through the benefit–risk ratio, as all calculated hazard quotients (HQ_EFA_) remained well below 1 (Table 3).

Across preparation methods, HQ_EFA_ values were consistently low, indicating negligible risk relative to the intake of 0.500 g EPA  +  DHA. Notably, Pb and Cd exhibited the highest values, particularly Pb in smoked bluefish, but these levels still fell below the threshold of concern [35]. Our findings are in line with the results reported by Gladyshev et al. (2020) [60] for smoked fish from Siberia, confirming that the intake of long-chain polyunsaturated fatty acids generally outweighs the potential risks from toxic elements.

Quantifying the benefit–risk ratio (LC-PUFA content to toxic/e elements) could provide a more accurate assessment of the nutritive value of seafood rather than using concentration values alone, especially when the concentration of toxic elements is below the maximum residue limits. The content of toxic and trace elements in cooked bluefish did not decrease the nutritional value in terms of n-3 LCPUFA content.

## 4. Conclusions

Pan-frying, grilling, and smoking of bluefish species resulted in cooked samples with increased elemental concentrations compared to the raw specimens. The extent of loss of water during cooking, largely influenced by fish size, was the primary factor determining the rise in metal levels. Grilling, despite being performed at comparable temperatures to pan-frying, caused less water loss and therefore yielded lower concentrations of metals. Variations in metal loss due to evaporation were minimal and did not significantly affect the overall results. The mineral content stayed within safe consumption limits across all treatments. The cooked species analyzed represent valuable sources of the essential elements Zn, Fe, and Cr, while the levels of the toxic Cd and Pb were within safe limits, posing no appreciable health risk to consumers.

The THQ and HI values for all elements were below the USEPA guideline, indicating no potential health risk from consuming bluefish from Bulgarian waters. TR values for carcinogenic metals also remained within acceptable limits (10^−6^–10^−4^). For consumers, the health benefits from the PUFA content in cooked fish surpassed the potential risks posed by toxic element levels. The results indicate that both raw and cooked bluefish are safe for consumption with respect to carcinogenic risk for an average adult consumer, assuming a daily intake of 17 g/day. However, this assessment may not fully apply to individuals belonging to high-consumption groups, who may require a more detailed risk evaluation. Cooking methods led to a significant increase in the concentrations of most elements compared to the raw samples. However, certain metals, such as Fe, showed decreased levels in this fish species following the proposed culinary treatments.

## Figures and Tables

**Figure 1 foods-15-00140-f001:**
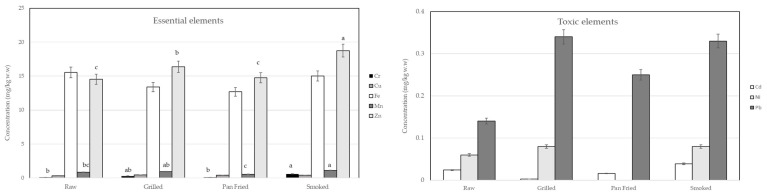
Mineral profile of raw and cooked bluefish (in mg/kg w.w; mean ± SD). The letters (a, b, c) mentioned in the bars indicate statistical difference (*p* < 0.05).

**Figure 2 foods-15-00140-f002:**
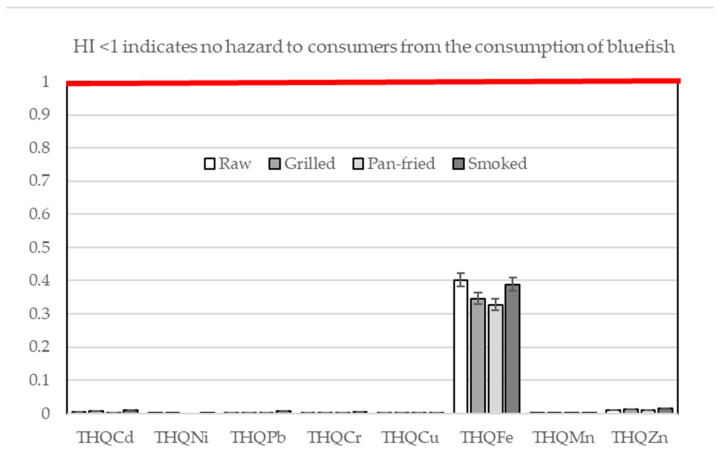
Non-carcinogenic risk (THQ: target hazard quotient) and hazard index (HI: hazard index) through the consumption of bluefish species.

**Table 1 foods-15-00140-t001:** Cooking parameters and equipment of bluefish (*Pomatomus saltatrix*, *L.*) samples.

Groups	Raw	Pan Fried	Grilled	Smoked
Cooking temperature	N/A	160 °C	60 °C	Pre-cooking: 40 °C Cooking: 90 °C Drying: 50 °C
Cooking time	N/A	5 min	4–6 min per side	Pre-cooking: 2 h 30 min Cooking: 8 h Drying: 2 h
Cooking methods and equipment	N/A	Stir-frying Wok	Blackstone^®^ Griddle (Logan, UT, USA)	Bradley^®^ Smoker (Delta, Canada)
Notes	N/A	Sunflower Oil	-	Charcoal

**Table 2 foods-15-00140-t002:** Estimated daily intake (EDI; mg/kg BW) values for each element analyzed for raw and cooked bluefish.

		Essential Elements					Toxic Elements		
		Cu	Fe	Zn	Cr	Mn	Cd	Ni	Pb
Estimated daily intake (mg/day/70 kg body weight)	Raw	8.0 × 10^−5^	3.8 × 10^−3^	3.5 × 10^−3^	2.2 × 10^−5^	2.0 × 10^−4^	5.8 × 10^−6^	1.5 × 10^−5^	3.4 × 10^−5^
	Grilled	1.1 × 10^−4^	3.2 × 10^−3^	4.0 × 10^−3^	7.0 × 10^−5^	2.2 × 10^−4^	7.0 × 10^−7^	1.9 × 10^−5^	8.3 × 10^−5^
	Pan Fried	9.5 × 10^−5^	3.1 × 10^−3^	3.6 × 10^−3^	2.4 × 10^−5^	1.3 × 10^−4^	3.9 × 10^−6^	n.d.	6.1 × 10^−5^
	Smoked	9.2 × 10^−5^	3.6 × 10^−3^	4.6 × 10^−3^	1.4 × 10^−5^	2.7 × 10^−4^	9.5 × 10^−6^	1.9 × 10^−5^	8.0 × 10^−5^
Intake		0.5 mg/kg b.w/day (PMTDI) [53]	0.8 mg/kg b.w/day * (PMTDI) [54]	0.3–1 mg/kg b.w/day (PMTDI) [55]	---	---	25 μg/kg b.w/month (PTMI) [55]	13 μg/kg b.w/day (TDI) [56]	0.00063 ** or/ 0.0015 *** (BMDL) [57]

PMTDI = provisional maximum tolerable daily intake; PTMI = provisional tolerable monthly intake; TDI = tolerable daily intake; BMDL = benchmark dose lower confidence limit. * For all sources except for iron oxide coloring agents, supplemental iron for pregnancy and lactation, and supplemental iron for specific clinical requirements; ** kidney effects (adults); *** cardiovascular effects (adults).

**Table 3 foods-15-00140-t003:** Hazard quotients (HQ_EFA_) for the benefit–risk ratio of essential fatty acids vs. elements via consumption of raw and cooked bluefish.

	Raw	Grilled	Pan-Fried	Smoked
DHA + EPA, mg/100 g EP *	959.07 ± 46.58	1478.40 ± 34.82	743.19 ± 59.74	2223.07 ± 161.91
Cu	0.002	0.005	0.009	0.003
Fe	0.006	0.002	0.004	0.002
Zn	0.011	0.010	0.015	0.009
Cr	0.002	0.005	0.009	0.003
Mn	0.012	0.009	0.011	0.007
Pb	0.005	0.012	0.012	0.023
Cd	0.018	0.014	0.014	0.013
Ni	0.000	0.001	N/A	0.001

* Values for DHA + EPA are reported in mg per 100 g edible portion (EP) of fish.

## Data Availability

The original contributions presented in this study are included in the article/Appendix A. Further inquiries can be directed to the corresponding author.

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
