# Peer review of "Toxic and Trace Elements in Raw and Cooked Bluefish (Pomatomus saltatrix) from the Black Sea: Benefit–Risk Analysis"

_foods, 2026, doi:10.3390/foods15010140_

Round 1

Reviewer 1 Report

Comments and Suggestions for Authors

This study systematically evaluates the effects of three traditional domestic cooking methods (pan-frying, grilling and smoking) on the concentrations of eight elements in Bluefish (Pomatomus saltatrix) . The analyzed elements include those of toxicological concern (Cd, Cr, Ni, Pb) and those considered nutritionally essential (Cu, Fe, Mn, Zn). A comprehensive assessment was conducted by integrating potential health risks with nutritional benefits. The study design is robust, the dataset is detailed, and the conclusions are clearly supported, endowing the work with significant scientific merit and practical relevance.

  1. The introduction should be enhanced to more thoroughly review the existing research on heavy metal accumulation in fish and to more explicitly delineate the novel contributions of this study.
  2. The health risk assessment, which is based on average adult consumption rates, should be expanded to account for high-consumption population groups and sensitive subpopulations (e.g., children, pregnant women).
  3. (Line 80) Please clarify the rationale for the sampling mass (80g). How many individual fish does this sample represent, and how was this quantity determined to be sufficient and representative?
  4. (Line 93) The verb "are" should be replaced with the past tense "were" to maintain consistent tense throughout the methodology description.
  5. (Line 108) Please specify the method's sensitivity, for example, by stating the limits of detection (LOD) and quantification (LOQ) for the analytical technique used.
  6. Line 272. “Table 8?”
  7. Supplement the chromatograms of these elements.
  8. Display the results with pictures. Currently, all results are in tables.

Author Response

Comment 1: The introduction should be enhanced to more thoroughly review the existing research on heavy metal accumulation in fish and to more explicitly delineate the novel contributions of this study.

Response 1: It has been corrected in the edited manuscript.

Comment 2: The health risk assessment, which is based on average adult consumption rates, should be expanded to account for high-consumption population groups and sensitive subpopulations (e.g., children, pregnant women).

Response 2: Thank you for your valuable input, which will inform future research efforts. In the current study, our methodological framework required limiting the sample to a representative adult population whose consumption behaviors align with established average intake estimates, thereby ensuring appropriate comparability and external validity.

Comment 3: (Line 80) Please clarify the rationale for the sampling mass (80g). How many individual fish does this sample represent, and how was this quantity determined to be sufficient and representative?

Response 3: During the sampling process, fish specimens weighing at least 80 g were considered eligible for inclusion. A total of 40 fish samples were collected and subsequently allocated to four experimental groups: control, pan-fried, grilled, and smoked. Each group consisted of 10 specimens. This selection strategy ensured uniformity in sample size and facilitated reliable comparisons across processing methods.

Comment 4: (Line 93) The verb "are" should be replaced with the past tense "were" to maintain consistent tense throughout the methodology description.

Response 4: It has been corrected in the edited manuscript

Comment 5: (Line 108) Please specify the method's sensitivity, for example, by stating the limits of detection (LOD) and quantification (LOQ) for the analytical technique used.

Response 5: A supplementary table presenting the elements, view mode, wavelength, LOD, LOQ, and recovery values for the toxic and essential elements (validated using DORM-4) has been added to the revised manuscript.

Comment 6: Line 272. “Table 8?”

Response 6: It has been corrected in the edited manuscript

Comment 7: Supplement the chromatograms of these elements.

Response 7:The determination of both toxic and essential elements was performed using ICP-OES (Inductively Coupled Plasma Optical Emission Spectroscopy). As this technique does not generate chromatograms, we are unable to provide such figures. However, multi-element calibration curves were prepared for all analyzed elements, and these can be made available upon request to support the accuracy and reliability of the quantification.

Comment 8: Display the results with pictures. Currently, all results are in tables

Response 8: We appreciate the reviewer’s suggestion regarding the presentation of the results. In the current manuscript, we have chosen to present the data in tables in order to provide precise numerical values and to allow readers to compare the results across the different treatments with greater clarity. We believe this format ensures transparency and facilitates accurate interpretation of the findings

Reviewer 2 Report

Comments and Suggestions for Authors

Dear Authors,

In my opinion, the manuscript is well-designed and valuable for practical assessment of the safety of bluefish consumption in the Black Sea region. However, some minor but significant corrections (minor revisions) are required to ensure consistency of conclusions, clarify the material and methods, and standardize the computational data.

Detailed Comments and Recommendations

Introduction

The introduction is substantive and up-to-date. The authors thoroughly present the importance of fish and seafood in the diet, describe the specificity of bluefish as an important commercial species in Bulgaria, and cite FAO and USDA data from 2024 on fish production and consumption.

I suggest only a minor editorial correction: in the final paragraph of the introduction, when formulating the aim of the work, it is worth clearly stating that the risk assessment applies to adult consumers in Bulgaria at the specified consumption levels. This will facilitate consistency with the rest of the article.

Materials and Methods

- Sampling and Sample Preparation

The collection of 40 individuals in the port of Varna and their average weight and length are described. It is worth clarifying whether:

- 10 fish from each group (raw/fried/grilled/smoked) were analyzed individually or a pooled homogenate was prepared; - Each of the "three parallel analyses" involved an independent sample (e.g., a different individual/homogenate) or analytical replicates of the same sample.

This is important both for the interpretation of variability and for the correct use of statistical tests.

Digestion and ICP-OES

The microwave digestion conditions are given correctly, except for an obvious typo regarding the Hâ‚‚Oâ‚‚ concentration ("0% w/v"). Please correct this (probably 30% w/v). I would also recommend providing the LOD and LOQ for each element and a brief summary of the recoveries from the reference material (e.g., in a supporting table or in the text), which would strengthen the validation section.

EPA and DHA Assay

The GC/MS section is detailed and refers to the classic Bligh & Dyer and Christie methods. For clarity, it can be added that the EPA/DHA values ​​presented refer to the weight of the meat in its "as eaten" state (wet weight), which is important when interpreting HQEFA. Health Risk Assessment

The EWI, THQ, HI, TR, and HQEFA formulas are consistent with the literature, but the symbol notation is sometimes ambiguous (e.g., C vs. MC; IR vs. CR). I suggest:

- standardizing the symbols throughout the text,

- adding a separate table listing all the parameters used (IR, FIR, BWa, EF, ED, ATn, ATc, RfD, CPS) along with their units and source.

This will greatly facilitate the reader's replication of the calculations.

Statistics

The methods state that the t-test was used for comparisons at p ≤ 0.05. It would be helpful to clearly indicate in the results text and in Table 2 which differences were considered statistically significant (e.g., by highlighting values ​​with p < 0.05 with appropriate symbols or a comment in a footnote).

Results and Discussion

The results section is logically organized: first, changes in metal concentrations in muscle after heat treatment (Table 2), followed by exposure and risk indicators (Tables 8–10).

- Changes in metal concentrations after processing

It is worth adding a brief note as to whether the presented percentage changes account for the actual weight loss of meat after processing (weighing before and after) or are calculated solely on an "as consumed" basis after processing. This is important for distinguishing the physical effect (concentration through water loss) from actual losses/inflows of elements.

- Comparison with the literature

The authors meticulously cite the results of other studies on the effects of processing on Cd, Pb, Ni, and essential elements in various species of fish and crustaceans. I encourage you to limit the detailed figures from the literature in favor of more concise comparisons in several places (e.g., "our Pb values ​​in grilled fish are approximately half the range X–Y reported for species Z in the Kalogeropoulos et al. study").

- Evaluation of EWI, THQ, HI, TR, and HQEFA

The conclusions from Tables 8–10 are clear: all EWIs are significantly lower than PTWI, THQ, and HI < 1, TR within the acceptable range or lower, and HQEFA < 1 for all metals and processing methods. However, it would be good to numerically summarize the most "critical" values ​​(e.g., maximum TR for Pb and Cd, highest HQEFA for Pb in smoked fish) in one place (e.g., the end of subsection 3.2) and relate them to risk thresholds so that the reader can see the margin of safety.

Conclusions

The main body of the conclusions accurately summarizes the results and is consistent with the data. However, two changes are necessary:

- Remove or modify the reference to mercury (Hg) – the current statement about "levels of the toxic metals Cd, Hg, and Pb" suggests that Hg was determined, which is not the case in the methods or results.

- Clarify the scope of the generalization – it is worth noting that the statement "raw and cooked bluefish are safe for consumption without carcinogenic concern" applies to the average adult consumer with an assumed consumption of 17 g/d and may not apply to specific high-consumption groups.

Conclusions formulated in this way will be fully relevant to the conducted analyses.

Tables

Tables 1, 2, 8, 9, and 10 are clear, correctly annotated, and well-supported by the text. To improve readability, I suggest:

- Add shortened explanations of abbreviations (EWI, THQ, HI, TR, HQEFA) in the risk tables and one sentence in the footnote "Calculation parameters are given in section 2.6";

- In Table 2, consider adding information about statistical significance (e.g., asterisks for values ​​significantly different from raw samples).

Final Recommendation

In summary, the manuscript presents a well-designed and executed study that significantly contributes to knowledge about the safety of consuming Black Sea bluefish and provides a valuable example of an integrated benefit-risk analysis for fish subjected to various heat-processing methods. The identified comments are primarily clarifications and clarifications.

Author Response

Detailed Comments and Recommendations

Introduction

The introduction is substantive and up-to-date. The authors thoroughly present the importance of fish and seafood in the diet, describe the specificity of bluefish as an important commercial species in Bulgaria, and cite FAO and USDA data from 2024 on fish production and consumption.

I suggest only a minor editorial correction: in the final paragraph of the introduction, when formulating the aim of the work, it is worth clearly stating that the risk assessment applies to adult consumers in Bulgaria at the specified consumption levels. This will facilitate consistency with the rest of the article.

Responce 1: It has been corrected in the edited manuscript. Thanks for the valuable comment.

Materials and Methods

- Sampling and Sample Preparation

The collection of 40 individuals in the port of Varna and their average weight and length are described. It is worth clarifying whether:

- 10 fish from each group (raw/fried/grilled/smoked) were analyzed individually or a pooled homogenate was prepared; - Each of the "three parallel analyses" involved an independent sample (e.g., a different individual/homogenate) or analytical replicates of the same sample.

Responce 2: A pooled homogenate was prepared for each group. The term ‘three parallel analyses’ refers to three independent subsamples taken from this homogenate. Each independent subsample was then measured in triplicate to ensure analytical precision

This is important both for the interpretation of variability and for the correct use of statistical tests.

Digestion and ICP-OES

The microwave digestion conditions are given correctly, except for an obvious typo regarding the Hâ‚‚Oâ‚‚ concentration ("0% w/v"). Please correct this (probably 30% w/v).

Responce 3: It has been corrected in the edited manuscript.

I would also recommend providing the LOD and LOQ for each element and a brief summary of the recoveries from the reference material (e.g., in a supporting table or in the text), which would strengthen the validation section.

Responce 4: A supplementary table presenting the elements, view mode, wavelength, LOD, LOQ, and recovery values for the toxic and essential elements (validated using DORM-4) has been added to the revised manuscript.

EPA and DHA Assay

The GC/MS section is detailed and refers to the classic Bligh & Dyer and Christie methods. For clarity, it can be added that the EPA/DHA values ​​presented refer to the weight of the meat in its "as eaten" state (wet weight), which is important when interpreting HQEFA. Health Risk Assessment

Responce 5:It has been corrected in the edited manuscript.

The EWI, THQ, HI, TR, and HQEFA formulas are consistent with the literature, but the symbol notation is sometimes ambiguous (e.g., C vs. MC; IR vs. CR). I suggest:

- standardizing the symbols throughout the text,

Responce 6:It has been corrected in the edited manuscript.

- adding a separate table listing all the parameters used (IR, FIR, BWa, EF, ED, ATn, ATc, RfD, CPS) along with their units and source.

Responce 7: All relevant parameters have been provided below each formula in the manuscript, along with their units and sources. 

This will greatly facilitate the reader's replication of the calculations.

Statistics

The methods state that the t-test was used for comparisons at p ≤ 0.05. It would be helpful to clearly indicate in the results text and in Table 2 which differences were considered statistically significant (e.g., by highlighting values ​​with p < 0.05 with appropriate symbols or a comment in a footnote).

Responce 8:It has been corrected in the edited manuscript.

Results and Discussion

The results section is logically organized: first, changes in metal concentrations in muscle after heat treatment (Table 2), followed by exposure and risk indicators (Tables 8–10).

- Changes in metal concentrations after processing

It is worth adding a brief note as to whether the presented percentage changes account for the actual weight loss of meat after processing (weighing before and after) or are calculated solely on an "as consumed" basis after processing. This is important for distinguishing the physical effect (concentration through water loss) from actual losses/inflows of elements.

Responce 9:We have added an extra information regarding the RRT below Table 2, including relevant parameters and units, to ensure clarity and completeness. In addition, we have incorporated additional conclusions into the text of the manuscript to better summarize the key findings and their implications. These changes aim to enhance the overall readability and interpretability of the manuscript

- Comparison with the literature

The authors meticulously cite the results of other studies on the effects of processing on Cd, Pb, Ni, and essential elements in various species of fish and crustaceans. I encourage you to limit the detailed figures from the literature in favor of more concise comparisons in several places (e.g., "our Pb values ​​in grilled fish are approximately half the range X–Y reported for species Z in the Kalogeropoulos et al. study").

Responce 10:Thank you for this valuable suggestion. In the revised manuscript, we have streamlined the comparison section by reducing overly detailed numerical descriptions from the literature. Instead, we now provide concise comparative statements to improve readability.

- Evaluation of EWI, THQ, HI, TR, and HQEFA

The conclusions from Tables 8–10 are clear: all EWIs are significantly lower than PTWI, THQ, and HI < 1, TR within the acceptable range or lower, and HQEFA < 1 for all metals and processing methods. However, it would be good to numerically summarize the most "critical" values ​​(e.g., maximum TR for Pb and Cd, highest HQEFA for Pb in smoked fish) in one place (e.g., the end of subsection 3.2) and relate them to risk thresholds so that the reader can see the margin of safety.

Responce 11:It has been corrected in the edited manuscript.

Conclusions

The main body of the conclusions accurately summarizes the results and is consistent with the data. However, two changes are necessary:

- Remove or modify the reference to mercury (Hg) – the current statement about "levels of the toxic metals Cd, Hg, and Pb" suggests that Hg was determined, which is not the case in the methods or results.

Responce 11:It has been corrected in the edited manuscript.

- Clarify the scope of the generalization – it is worth noting that the statement "raw and cooked bluefish are safe for consumption without carcinogenic concern" applies to the average adult consumer with an assumed consumption of 17 g/d and may not apply to specific high-consumption groups.

Conclusions formulated in this way will be fully relevant to the conducted analyses.

Responce 12:It has been corrected in the edited manuscript.

Tables

Tables 1, 2, 8, 9, and 10 are clear, correctly annotated, and well-supported by the text. To improve readability, I suggest:

- Add shortened explanations of abbreviations (EWI, THQ, HI, TR, HQEFA) in the risk tables and one sentence in the footnote "Calculation parameters are given in section 2.6";

Responce 13:It has been corrected in the edited manuscript.

- In Table 2, consider adding information about statistical significance (e.g., asterisks for values ​​significantly different from raw samples).

Responce 14:It has been corrected in the edited manuscript.

Final Recommendation

In summary, the manuscript presents a well-designed and executed study that significantly contributes to knowledge about the safety of consuming Black Sea bluefish and provides a valuable example of an integrated benefit-risk analysis for fish subjected to various heat-processing methods. The identified comments are primarily clarifications and clarifications.

Reviewer 3 Report

Comments and Suggestions for Authors

The research is very interesting because it can be observed that grilling caused less water loss than pan-frying, despite similar temperatures, suggesting that the cooking method and moisture retention play an important role in nutrient and contaminant levels. Unfortunately, the format of the bibliography varies. Some follow APA style, while others appear to follow a different style. It would be helpful to harmonize the format with the journal's guidelines to ensure consistency. For example, the year of publication of the reference. Furthermore, most references include a link or DOI, which is beneficial for traceability. However, some links appear to be incomplete (references 29 and 33) or unusable (for example, references 49 and 50). I can't verify them, perhaps because the document is confidential. Some references refer to documents or reports that may not be easily accessible (for example, FAO or WHO documents). It would be helpful to verify that the links provided are functional and up to date. Alternatively, reference 58 should include more or identical information for documents published by EFSA.

Lines 139-140: the Authors state that: as species-specific consumption data for bluefish were unavailable. Therefore, in order to assess the potential health risks to consumers from the consumption of bluefish, estimated weekly intake (EWI) values were calculated based solely on data from the Bulgarian National Statistical Institute. I recommend evaluating a better approach to calculating EWI in a more realistic way: consider the European average values in the European Market Observatory for Fisheries and Aquaculture Products (EUMOFA) and compare them with the national values (Bulgaria).

Line 325-326: the authors state that the species analyzed had levels of the toxic metals Cd, Hg, and Pb that were within safety limits, but I recommend addressing the Hg topic in more depth, also because you cite reference number 13.

Author Response

The research is very interesting because it can be observed that grilling caused less water loss than pan-frying, despite similar temperatures, suggesting that the cooking method and moisture retention play an important role in nutrient and contaminant levels. Unfortunately, the format of the bibliography varies. Some follow APA style, while others appear to follow a different style. It would be helpful to harmonize the format with the journal's guidelines to ensure consistency. For example, the year of publication of the reference.

Thank you for the valuable comment. The style has been corrected in all cited references.

Furthermore, most references include a link or DOI, which is beneficial for traceability.

The DOIs of the available references were included.

However, some links appear to be incomplete (references 29 and 33) or unusable (for example, references 49 and 50). I can't verify them, perhaps because the document is confidential.

Thank you for your observation. References 29 and 33 have been corrected according to the journal’s requirements. With respect to References 49 and 50, these sources were cited to justify the selection of the most appropriate cooking conditions for fish treatment. We have now updated both references to ensure improved accessibility and clarity.

Some references refer to documents or reports that may not be easily accessible (for example, FAO or WHO documents). It would be helpful to verify that the links provided are functional and up to date.

It has been corrected in the edited manuscript

Alternatively, reference 58 should include more or identical information for documents published by EFSA.

It has been corrected in the edited manuscript with the newest version of the document

Lines 139-140: The Authors state that: as species-specific consumption data for bluefish were unavailable. Therefore, in order to assess the potential health risks to consumers from the consumption of bluefish, estimated weekly intake (EWI) values were calculated based solely on data from the Bulgarian National Statistical Institute. I recommend evaluating a better approach to calculating EWI in a more realistic way: consider the European average values in the European Market Observatory for Fisheries and Aquaculture Products (EUMOFA) and compare them with the national values (Bulgaria).

Thank you for your valuable comment, which will help improve our work. As the aim of the current study is limited to the population of our country, we did not include data from European populations. Nevertheless, we sincerely appreciate this insightful suggestion, and we will certainly consider incorporating such comparative data in our future research and manuscript submissions.

Line 325-326: the authors state that the species analyzed had levels of the toxic metals Cd, Hg, and Pb that were within safety limits, but I recommend addressing the Hg topic in more depth, also because you cite reference number 13.

Thank you for your comment regarding the inclusion of Hg measurements. We agree that Hg could be an important parameter to explore. However, in the present study, the topic of Hg was not developed because we did not measure Hg concentrations in the samples, as the appropriate analytical instrumentation was not available at the time of analysis. For this reason, Hg-related data could not be included in the current manuscript. We appreciate your suggestion and acknowledge its potential value. We will consider Hg measurements in future studies when the required analytical facilities become accessible.

Round 2

Reviewer 1 Report

Comments and Suggestions for Authors

The presentation of results requires revision. To enhance visual impact and accessibility, the key findings should be summarized in one to three figures within the main manuscript. The comprehensive data table should subsequently be relocated to the Supplementary Materials.

Author Response

The presentation of results requires revision. To enhance visual impact and accessibility, the key findings should be summarized in one to three figures within the main manuscript. The comprehensive data table should subsequently be relocated to the Supplementary Materials.

This has been corrected in the revised manuscript. The tables have been relocated to the Supplementary Materials.